# DHA Attenuates Cerebral Edema Following Traumatic Brain Injury via the Reduction in Blood–Brain Barrier Permeability

**DOI:** 10.3390/ijms21176291

**Published:** 2020-08-31

**Authors:** Zhuo-Hao Liu, Nan-Yu Chen, Po-hsun Tu, Chen-Te Wu, Shao-Chieh Chiu, Ying-Cheng Huang, Siew-Na Lim, Ping K. Yip

**Affiliations:** 1Department of Neurosurgery, Chang Gung Memorial Hospital at Linkou, Chang Gung Medical College and University, Taoyuan County 333, Taiwan; albert3343@gmail.com (P.-h.T.); ns3068@gmail.com (Y.-C.H.); 2Department of Internal Medicine, Chang Gung Memorial Hospital at Linkou, Chang Gung Medical College and University, Taoyuan County 333, Taiwan; nellychen@gmail.com; 3Department of Medical Imaging and Intervention, Chang Gung Memorial Hospital at Linkou, Chang Gung Medical College and University, Taoyuan County 333, Taiwan; melik@adm.cgmh.org.tw; 4Center for Advanced Molecular Imaging and Translation, Chang Gung Memorial Hospital at Linkou, Taoyuan County 333, Taiwan; jeff17342001@gmail.com; 5Department of Neurology, Chang Gung Memorial Hospital at Linkou, Chang Gung Medical College and University, Taoyuan County 333, Taiwan; siewna.lim@gmail.com; 6Queen Mary University of London, Barts and The London School of Medicine and Dentistry, Blizard Institute, Centre for Neuroscience, Surgery & Trauma, London E1 2AT, UK

**Keywords:** traumatic brain injury, docosahexaenoic acid, edema, blood–brain barrier, aquaporin 4, metalloproteinase 9

## Abstract

Traumatic brain injury (TBI) could result in edema and cause an increase in intracranial pressure of the brain resulting in mortality and morbidity. Although there is hyperosmolarity therapy available for this pathophysiological event, it remains controversial. Recently, several groups have shown docosahexaenoic acid (DHA) to improve functional and histological outcomes following brain injury based on reduction of neuroinflammation and apoptosis. However, the effect of DHA on blood–brain barrier (BBB) dysfunction after brain injury has not been fully studied. Here, a controlled cortical impact rat model was used to test the effect of a single dose of DHA administered 30 min post injury. Modified neurological severity score (mNSS) and forelimb asymmetry were used to determine the functional outcomes. Neuroimaging and histology were used to characterize the edema and BBB dysfunction. The study showed that DHA-treated TBI rats had better mNSS and forelimb asymmetry score than vehicle-treated TBI rats. Temporal analysis of edema using MRI revealed a significant reduction in edema level with DHA treatment compared to vehicle in TBI rats. Histological analysis using immunoglobulin G (IgG) extravasation showed that there was less extravasation, which corresponded with a reduction in aquaporin 4 and astrocytic metalloprotease 9 expression, and greater endothelial occludin expression in the peri-contusional site of the TBI rat brain treated with DHA in comparison to vehicle treatment. In conclusion, the study shows that DHA can exert its functional improvement by prevention of the edema formation via prevention of BBB dysfunction after TBI.

## 1. Introduction

Traumatic brain injury (TBI) can affect people of all ages and is a major cause of death and disability, with an incidence of approximately 10 million people worldwide [1]. In particular, TBI causes more death in males at the age of under 35 years old than all other disease combined [2]. To date, there is only standard medical intervention and care as no effective therapy for TBI is available [3]. However, there are recent pre-clinical studies to suggest promising treatments for TBI. One example is the use of omega-3 polyunsaturated fatty acid that is essential in normal brain development involved in cognition and other brain functions [4].

Recent studies have demonstrated that dietary depletion of omega-3 fatty acid during gestation and postnatal development can induce cognitive deficits and exacerbated neuronal death after TBI [5]. In a more therapeutically applicable scenario, a study that had provided TBI rats with dietary supplementation for 1 month after injury showed decreased TBI-induced axonal damage and apoptosis [6]. Another study that injected omega-3 fatty acid intraperitoneally 30 min post injury and then daily for 1 week demonstrated an attenuation of TBI-induced neuronal apoptosis [7]. Given that omega 3 fatty acid consists mainly of eicosapentaenoic acid (EPA) and docosahexaenoic acid (DHA), it is important to understand the role of EPA and DHA individually. Interestingly, EPA has not been a promising agent in CNS diseases. A spinal cord injury study in rodents showed that DHA but not EPA can reduce early inflammatory response [8]. Furthermore, EPA has shown to accelerate the disease progression of amyotrophic lateral sclerosis in a rodent model [9].

Treatment with DHA alone has shown beneficial outcomes after TBI. A recent study showed that intragastric treatment with 740 mg/kg/day DHA at 30 min post injury then daily for 15 days protected against TBI-induced motor and cognitive deficits, and apoptosis [10]. Another study that injected 16 mg/kg DHA at 30 min and subsequent doses at 1, 3 and 5 days post injury showed improved motor and cognitive deficits induced by the TBI [11]. Interestingly, a recent study demonstrated that a single 500 nmol/kg dose of DHA injected intravenously can provide a reduction in TBI-induced lesion size, neuroinflammation, and axonal injury [12]. Although several studies have indicated the mechanism of action of DHA on TBI recovery is via reduction in neuroinflammation and apoptosis, other pathophysiological events can also occur after TBI.

In TBI, the primary injury is quickly followed by the secondary injury that involves sequelae including neuroinflammation, oxidative stress, apoptosis, and neurovascular dysfunction [13]. TBI-induced blood–brain barrier (BBB) disruption can increase cerebral vascular permeability, leading to the formation of brain edema [14]. Clinically, peri-hemorrhagic edema is commonly observed in TBI patients and causes elevations in intracranial pressure (ICP), hydrocephalus, or brain herniation. Raised ICP is associated with poor outcome as it can cause permanent brain damage and death [15]. Although hyperosmolar therapy using hypertonic saline or mannitol is available, the different hyperosmolar agent type and the use of various doses cause controversy regarding the effectiveness and safety of this procedure [16,17,18].

Given the importance of TBI-induced edema, the present study was designed to determine whether DHA can reduce edema and protect against BBB dysfunction after TBI in rats. Interestingly, aquaporin 4 (AQP4) and matrix metalloproteinase (MMP9) are integral components of the pathophysiology of brain edema and have become viable candidates for potential therapeutic targets for DHA [19,20]. We reported here that a single delayed DHA treatment can reduce brain edema following TBI via neuroimaging and histological analysis of AQP4, MMP9 and occludin.

## 2. Results

### 2.1. DHA Reduces Sensorimotor Deficits after TBI

To assess sensorimotor function, the mNSS and cylinder tests, which have been widely used in experimental models of stroke and TBI were carried out [21,22]. There was a significant difference in the mNSS score (mean ± SEM) between the DHA-treated (7.9 ± 0.3) and vehicle-treated TBI rats (7.9 ± 0.1) in comparison to the sham-operated rats (0.7 ± 0.7) at 1 day post injury (Figure 1A). Interestingly, from day 2 onwards, the DHA-treated rats expressed a greater rate of recovery compared to the vehicle-treated TBI group (Figure 1A). Between day 6 and 7 post injury, the mNSS score for DHA-treated TBI rats (day 6, 1.9 ± 0.3; day 7, 1.7 ± 0.2) were significantly lower than the vehicle-treated TBI rats (day 6, 3.3 ± 0.2; day 7, 3.1 ± 0.1) (Figure 1A).

The cylinder task is primarily a test of somatosensory neglect of forelimb [23]. In this study, all animals showed similar forelimb use asymmetry scores between −0.3 to 0.05 in rats 1 day prior to receiving any brain injury (Figure 1B). After TBI, there was an increase in forelimb use asymmetry score, but no significant difference in forelimb use asymmetry score (mean ± SEM) between the DHA- (2.7 ± 1.9) and vehicle-treated (4.6 ± 1.1) TBI rats at 1 day post injury (Figure 1B). However, at day 2 post injury, the DHA-treated rats (3.3 ± 1.6) were significantly different from the vehicle-treated rats (7.5 ± 0.8). The significant difference between DHA-treated and vehicle-treated TBI rats remained significantly different at day 3 and 4 post injury (Figure 1B). However, the significant difference between the two TBI groups was diminished by day 5 post injury. The sham-operated rats expressed a forelimb use asymmetry score (mean ± SEM) between the highest (0.8 ± 0.1) at day 2 and lowest forelimb asymmetry score (0.3 ± 0.1) at day 3 post surgery (Figure 1B).

Overall, TBI rats treated with DHA recovered significantly quicker in motor function than the vehicle-treated TBI group from day 2 onward.

### 2.2. DHA Attenuates Cerebral Edema after TBI

To assess the TBI-induced edema temporally, MRI examinations were carried out. This technique has been used in the clinical setting to quantify the severity of contusion edema following brain injury. As expected, T2-weighted imaging (T2WI) revealed no lesion in sham-operated rats (Figure 2A–C), but lesions involving the cortical and subcortical areas of the brain in DHA- and vehicle-treated TBI rats (Figure 2D–I) were observed. In comparison to sham-operated group, the edematous volume computed from T2WI was significantly increased in rats with TBI, in 1, 3, and 7 days after brain contusion (Figure 2J). However, DHA-treated rats had a significant reduction in edematous brain tissue size in comparison to the vehicle-treated TBI rats at 1, 3, and 7 days post injury (Figure 2J).

Overall, TBI rats treated with DHA had significantly smaller edema at the impact site than the vehicle-treated TBI group from day 1 onward.

### 2.3. DHA Attenuates Extravasation after TBI

The cause of edema is the extravasation of fluid into the cerebral parenchyma due to blood–brain barrier (BBB) dysfunction [24]. To examine the extravasation level, the expression of endogenous immunoglobulin G (IgG) within the cerebral parenchyma at the peri-contusion site at 7 days post injury was analyzed [25]. There was limited IgG expression (mean ± SEM) in sham-operated rats (7.0 ± 0.6) (Figure 3A). However, Immunostaining for IgG revealed a significant increase in IgG expression after TBI injury (Figure 3B–D). In comparison between the two TBI groups, the DHA-treated rats (11.1 ± 0.7) had significantly less IgG expression compared to the vehicle-treated rats (14.2 ± 0.8) (Figure 3D).

Overall, TBI rats treated with DHA have significantly less extravasation at the peri-contusion site than the vehicle-treated TBI group at day 7 post injury.

### 2.4. DHA Alleviates BBB Damage after TBI

BBB dysfunction could be due to endothelial cell loss, which can be identified using the endothelial cell marker RECA-1 using immunohistochemistry [26]. In sham-operated rats, RECA-1 expression (mean ± SEM) was (28.8 ± 0.5), and that RECA-1 immunostaining showed the presence of many large and small caliber cerebral blood vessels in the cortex (Figure 4A,D). In contrast, for rats that had received a TBI and received vehicle treatment, the RECA-1 expression was (25.1 ± 0.3), and predominantly fragmented and small caliber RECA-1 immunopositive cerebral blood vessels in the pericontusional region were observed (Figure 4B,D). However, in DHA-treated TBI rats, the RECA-1 expression was (27.5 ± 0.3), which was significantly higher than the vehicle-treated TBI rats, and longer strands of cerebral blood vessels were observed (Figure 4C,D).

Overall, TBI rats treated with DHA had significantly less endothelial cell loss at the peri-contusion site than the vehicle-treated TBI group at day 7 post injury.

### 2.5. DHA Reduces Aquaporin4 Expression after TBI

Apart from endothelial cell loss, BBB dysfunction could be due to abnormal tight junction formation on the cerebral vasculature by the astrocytic endfeet [27]. One molecule found highly expressed in astrocytic endfeet is aquaporin4 (AQP4) [28]. In a study using AQP4 knock out mice, they showed reduced edema from a model of acute vasogenic brain edema and middle cerebral artery occlusion [29]. Irrespective of the treatment or injury, the expression of AQP4 was predominantly on cerebral blood vessels (Figure 5A–C). In sham-operated rats, AQP4 expression (mean ± SEM) was (13.5 ± 0.4), (Figure 5A,D). In contrast, the AQP4 expression in vehicle-treated TBI rats (27.0 ± 1.1) was significantly higher in the pericontusional region than the sham-operated rats (Figure 5B,D). In TBI rats treated with DHA, the AQP4 expression was (22.3 ± 0.8), which although it was significantly higher than the sham-operated rats, was significantly lower than the vehicle-treated TBI rats (Figure 5C,D).

Overall, TBI rats treated with DHA had significantly less AQP4 expression at the peri-contusion site than the vehicle-treated TBI group at day 7 post injury.

### 2.6. DHA Reduces Astrocytic MMP9 Expression after TBI

BBB dysfunction from CNS insult has been associated with the activation of astrocyte via the upregulation of glial fibrillary acidic protein (GFAP) [27]. In this study, an increase in GFAP expression was also observed in TBI rats in comparison to the sham-operated rats (Figure 6A–C). Although there was a significant difference between the vehicle-treated TBI rats, compared to the sham-operated rats, there was no significant difference between sham-operated or vehicle-treated TBI rats to DHA-treated TBI rats at 7 days post injury (Figure 6D).

Interestingly, astrocytes have a central role in the release of matrix metalloproteinases (MMP), such as MMP9 in ischemic injury [30]. Since MMP9 has been implicated in BBB dysfunction via increase BBB permeability, it would be important to study whether astrocytic MMP9 expression was altered with DHA treatment after TBI. Dual immunostaining for GFAP and MMP9 showed co-localization of MMP9 with the GFAP immunopositive astrocytes surrounding cerebral blood vessels (Figure 6E–G). In sham-operated rats, astrocytic MMP-9 expression (mean ± SEM) was (20.1 ± 0.4) in the cortex (Figure 6E, H). In contrast, rats that have received a TBI and vehicle treatment, the MMP9 expression was (34.4 ± 2.4) in the pericontusional region (Figure 6F,H). However, in DHA-treated TBI rats, the MMP9 expression was (23.9 ± 0.2), which was significantly lower than the vehicle-treated TBI rats (Figure 6G,H).

Overall, TBI rats treated with DHA have significantly less astrocytic MMP9 expression around the cerebral blood vessels at the peri-contusion site than the vehicle-treated TBI group at day 7 post injury.

### 2.7. DHA Increases Endothelial Occludin Expression after TBI

To further investigate the BBB dysfunction from CNS insult, the tight junction marker occludin and the endothelial cell marker CD31 were analyzed to test if there was an increase in leakiness due to reduction in tight junction integrity within endothelial cells after TBI. Analysis of the data showed a non-significant decrease in CD31 expression in rat brains of vehicle-treated TBI compared to sham-operated rats, but it was significantly lower than in TBI rats treated with DHA (Figure 7A–F,J). Dual immunostaining for the tight junction marker occludin and CD31 showed that there was a significant reduction in overall occludin expression (Figure 7A–I,K) and occludin expression within CD31 immunopositive cells (Figure 7A–I,L) in rat brains after TBI compared to in sham-operated rats or TBI rats post-treated with DHA.

Overall, these data suggest that the BBB dysfunction was due to a significant reduction in tight junction integrity within endothelial cells at the peri-contusion site after TBI at day 7 post injury, which was reversed with DHA treatment.

## 3. Discussion

Several recent independent studies have shown that DHA promotes beneficial effects in TBI, resulting in improved functional outcomes, and reduced apoptosis and neuroinflammation [10,11,12]. However, the mechanism underlying these functional changes, in particular with edema, remain poorly understood. The present study demonstrated that a single intravenous bolus of DHA injected 30 min after the CCI injury can provide improved functional recovery in the mNSS and forelimb use asymmetry score. Neuroimaging using MRI can confirm an immediate edema reduction at the contusion site in TBI rats treated with DHA. Histological analysis revealed DHA-treated TBI rats had reduced extravasation of IgG, which correlated with more intact blood vessels immunostained with RECA-1, reduced AQP4 expression, reduced astrocytic MMP9 expression, and increased endothelial occludin expression. Therefore, this study is the first to report the reduction of post-traumatic brain edematous change after acute DHA administration and provide a mechanism of action.

The mechanism of DHA on improved functional outcomes after TBI has been suggested by other research groups and is predominantly involved in neuroinflammation and apoptosis. For example, Zhu and colleagues proposed that in their rat fluid percussion injury model that DHA activates the nuclear factor E2-related factor 2-antioxidant response element (Nrf2-ARE) signaling pathway [31]. A possible role for Nrf2-ARE in TBI is to provide neuroprotectivity via increasing Bcl2, reducing Bax, and cleavage of caspase 3 [31]. Another research group, which focused on the neuroprotective aspect, suggested in their rat controlled cortical impact model that DHA suppresses the TLR4/NF-κB signaling pathway [11]. Suppression of TLR4 expression and inhibition of inflammatory mediators such as NF-κB leads to suppression of neuroinflammation. A recent study by Thau-Zuchman and colleagues suggested in their mouse controlled cortical impact model that DHA provides neuroprotection via the upregulation of neuroprotective mediators such as DHA-derived resolvins and protectins, and reduction of neuroinflammatory response [12].

In this study, the focus on the mechanism of action of improved functional outcome by DHA was on edema as a breakdown of BBB is one of the main contributors interfering brain recovery from secondary damage [32]. Following trauma insult to the brain, the normal BBB formed by tightly connected neurovascular unit incurs a concomitant shear injury, resulting in impaired BBB regulation, which in turn, leads to brain edema [33]. Experimental TBI models have shown that BBB dysfunction is associated with damage to endothelial cells [26,34]. In this study, a decrease in the endothelial cell marker RECA-1 was observed after TBI. Similarly, other TBI studies involving fluid percussion injury and blast injury have shown microvascular loss via a reduction of RECA-1 immunostaining [35,36,37] and microvascular basal lamina damage [38]. As aforementioned, TBI rats treated with DHA in this study revealed a significant reversal in the decrease in RECA-1 expression at the peri-contusion site. Not surprisingly, it has been shown that DHA reduces TNF-α-induced endothelial cell injury by inhibiting gene expression related to endothelial dysfunction, such as plasminogen activator inhibitor 1 (SERPINE1/PAI-1) and lectin-like oxidized low-density lipoprotein receptor-1 (LOX1) [39]. It has been shown that PAI-1 and LOX1 contribute to vascular endothelial dysfunction [40,41]. However, it should be noted that the reduction in RECA1+ blood vessels after TBI may not be due to endothelial cell loss but a reduction in RECA1 expression. This is supported by the non-significant decrease in the endothelial CD31 expression after TBI compared to sham (Figure 7J).

There is compelling evidence that the upregulation of AQP4 mediates BBB disruption in repetitive ethanol intoxication [42,43], stroke [44] and TBI [45,46]. AQP4 is abundantly expressed in astrocyte endfeet processes surrounding blood vessels that play key roles in basal membrane and tight junction protein degradation, and state modulation [47]. Increased pericontusional expression of AQP4 correlates with the development of cellular edema after TBI in humans and rodents [46,48]. This protein enhances edema formation that subsequently contributes to an elevated ICP, thus causing cell death in brain tissues [28]. Our results suggest that the DHA-induced reduction of BBB dysfunction is associated with the down-regulation of AQP4. Although the exact mechanism of DHA-conferred BBB protection via AQP4 is unclear, it has been shown that pro-inflammatory cytokines such as TNF-α which can increase astrocytic AQP4 levels [42] are reduced by DHA [49]. Therefore, one potentially important mechanism whereby DHA provide functional improvements against TBI may be the modulation of central AQP4 production and activity.

Another molecule related to BBB dysfunction is MMP9, which is up-regulated after stroke [50] and TBI [51,52,53]. Furthermore, MMP9 appears to contribute to both neuronal cell loss following TBI and the facilitation of regenerative processes [54,55] Interestingly, clinical studies have found that MMP9 in cerebrospinal fluid was elevated in stroke and TBI patients [56,57]. In this study, the expression of MMP9 was predominantly within GFAP immunopositive astrocytes surrounding blood vessels. Increased MMP9 around blood vessels has been suggested to be pathologically involved in the disruption of the parenchymal basement membranes, so enabling an increase in ICP and the entering of peripheral immune cells into the brain parenchyma [58,59]. A recent study showed increased MMP9 contributes to brain extravasation and the onset of coma, which can be blocked with an MMP9 monoclonal antibody in an acute liver failure mouse model [60]. Furthermore, MMP9 secretion can cause demyelination and axonal injury, in which axonal damage is considered to be a major cause of secondary injury [61,62]. The reduction of astrocytic MMP9 expression could ameliorate the BBB breakdown, which can reduce the edematous change after brain injury.

Although exactly how DHA reduces MMP9 to provide BBB protection after TBI is unknown, a recent in vitro study demonstrated that DHA could inhibit the expression of MMP9 mRNA and protein expression via modulation of the MAPK signaling pathway and PPAR-γ/NF-κB activity [63,64]. In addition, one recent study demonstrated that DHA can mitigate BBB disruption by reducing the MMP2 and 9 activation after hypoxic–ischemic brain injury [65]. Therefore, given that MMPs when activated can cause BBB dysfunction by degrading the tight junctions and basal lamina proteins, leading to BBB leakage and brain edema in various diseases [66,67,68], reducing MMP9 with DHA and improving tight junction integrity as demonstrated in this study would be effective to treating TBI. Although we have demonstrated that DHA can affect the paracellular permeability (i.e., tight junction integrity), the transcellular permeability (i.e., increase in transcytosis) could also be affected. A recent study showed that DHA is transported into the brain via the lipid transporter Mfsd2a located at the CNS luminal plasma membrane endothelial cell, a form of caveolae-mediated transcytosis [69].

Apart from the effects of DHA on MMP9, DHA can also reduce endothelial cell injury by reducing the inflammatory response. In particular, DHA has been shown to prevent tumor necrosis factor alpha-induced endothelial dysfunction [39]. Furthermore, DHA can attenuate vascular cell adhesion molecule-1 (VCAM-1) expression and reduce the activation of the transcription factor NF-κB in pro-inflammatory TNFα-treated human endothelial cells [70]. Since DHA is found primarily within neuronal tissue of the CNS and retina [71], one may query how DHA can affect endothelial cells. One potential method is that DHA administered systemically can diffuse directly into cells or interact with fatty-acid-transport proteins (FABPs), which can facilitate the transfer of fatty acids across extra- and intra-cellular membranes of cells such as endothelial cells that line the blood vessels [72]. Among the nine members of FABPs, brain FABP (B-FABP) is distinguished from other FABPs by its strong affinity for omega PUFAs, in particular DHA, which indicates that B-FABP may be intricately linked to the role of DHA in the nervous system [73,74].

In summary, our present study suggests that a 30 min delay in acute systemic administration of DHA following TBI can reduce cerebral edema, which enables a decrease in the deficits of neurologic function by ameliorating the BBB dysfunction via suppression of AQP4 and MMP9 and limiting occludin loss at the pericontusion site. Therefore, this study provides a novel mechanism for the neuroprotective role of DHA in TBI and suggests that DHA may represent a clinically safe AQP4 and MMP9 suppressor.

## 4. Materials and Methods

### 4.1. Surgical Procedure

Young adult male Sprague–Dawley rats (200–225 g, *n* = 7–8 per group) were used for this experiment according to the Institutional Animal Care and Use Committee at Chang Gung Memorial Hospital. Rats were anesthetized with isoflurane and then secured to a stereotaxic frame. After a longitudinal midline incision of the scalp, the skin was retracted and the skull was exposed. A 4-mm-diameter circular craniotomy was made over the left sensorimotor cortex, centered at the bregma and 3.5 mm lateral to the midline. A controlled-cortical impact (CCI) injury model was created with a unilateral cortical contusion injury to the left hemisphere. The trauma to the brain was produced using a 2.5 mm diameter flat-tipped metal impactor attached to a pneumatically controlled piston driven by a computer (Pinpoint Precision cortical impactor, Hatteras Instruments, Cary, NC, USA) on a stereotaxic micromanipulator. The piston was angled at 22.5° from vertical to make contact perpendicularly to the surface of the brain. Impact velocity of 3 m/s and a deformation depth below the dura of 2 mm was used to produce a pericontusional region over the forelimb cortex, leaving the underlying corpus callosum anatomically intact but deficient in it is connectivity. After the brain trauma, the bone flap was replaced and sealed in place with bone cement, and the skin incision was closed with sutures. The sham-operated rats were treated exactly as the TBI rats, but without the brain exposed to the mechanical trauma. All animals received the post-operative care, which includes subcutaneous injection of analgesics (Buprenorphine, 0.01 mg/kg body weight) and normal saline (2mL) for 3 days following surgery.

### 4.2. Drug Administration Procedure

A single administration of the either vehicle (0.2% ethanol in saline) or DHA (Sigma D2534, 250 nmoL/kg) in a volume of 5 mL/kg was carried out intravenously 30 min after the trauma to the brain. The DHA dose chosen was based on a previous study showing DHA induced functional improvements in a rat spinal cord injury model [75].

### 4.3. Modified Neurological Severity Scores

Each animal’s neurological function was evaluated by a set of modified Neurological Severity Scores (mNSS) by an investigator who was blinded to the experimental groups. The mNSS includes a composite of motor (muscle status and abnormal movement), sensory (visual, tactile and proprioceptive), reflex and balance tests according to previously described (Table 1) [76,77]. One point was allocated for the inability to perform each test while zero point was allocated for the ability to perform the task. An overall composite score was provided to determine the impairment of neurological function was graded on a scale of 0 to 18 (normal score, 0; maximum deficit score, 18). Therefore, the higher the score, the more severely injured the brain was from the trauma.

### 4.4. Forelimb Asymmetry Test

Forelimb use during exploratory activity was analyzed by videotaping rats in a transparent cylinder (20 cm diameter and 30 cm height) for 3 to 10 min depending on the degree of activity during the trial. A mirror was placed to the side of the cylinder at an angle to enable the recording of forelimb movements even when the animal was facing away from the camera. Scoring was done by an experimenter blinded to the experimental group using a video cassette recorder with slow-motion and clear stop frame capabilities. The behavior was scored according to the following criteria: (1) independent use of the left or right forelimb contacting the wall of the cylinder during a full rear to initiate a weight-shifting movement or to regain center of gravity while moving laterally in a vertical posture, and (2) simultaneous use of both the left and right forelimbs for contacting the cylinder wall during a full rear.

Behavioral data analysis was quantified by determining the occasions when the unimpaired left forelimb was used as a percentage of total number of forelimbs used (U); the occasions when the impaired right forelimb was used as a percentage of total number of forelimb used (I); and the occasions when both forelimbs were used simultaneously (or nearly simultaneously during lateral side-stepping movements) as a percentage of total number of forelimb used (B). A single overall limb use asymmetry score was calculated as follows: Forelimb use asymmetry score = [U/(U + I + B)] − [I/(U + I + B)].

### 4.5. MRI Image Evaluation

The MRI experiments were performed with a 3.0T GE CSI system equipped with shielded gradients capable of producing 20 G/cm. The rat was anesthetized with isoflurane and positioned supine with the head inside a 5.5 cm diameter bird cage radiofrequency coil. To monitor brain edema evolution, T2-weighted images were acquired at each imaging time point at 1, 3, and 7 days after brain injury. Brain edematous change was quantified by measuring the T2-hyperintense area at the level of bregma 0.48 mm.

### 4.6. Immunohistochemical Staining

At the designated time point, rats were deeply anesthetized and then transcardially perfused with 0.9% saline followed with 4% paraformaldehyde (PFA) at 7 days following CCI injury. The brains were removed from animals and post-fixed overnight in fresh 4% PFA, then cryoprotected in 20% sucrose for at least 2 days. To investigate the change in cortical neurons in the motor cortex after SCI, coronal sections of the rat brains were cut in a rostrocaudal manner using a cryostat and stored at −20 °C until further processed. Randomly selected slides containing the brain sections were removed from −20 °C and then washed with gentle agitation in phosphate-buffered saline (PBS) (3 × 5 min). Thereafter, the sections were incubated in 10% normal donkey or goat serum for 30 min followed by overnight incubation with immunoglobulin G (IgG, 1:500, Vector Laboratories Burlingame, CA, USA), anti-rat endothelial cell antigen-1 (RECA-1, 1:250, Bio-Rad, Hercules, CA, USA), NeuN (1:1000, Merck Millipore, Burlington, MA, USA), AQP4 (1:1000, Santa Cruz Biotechnology, Dallas, TX, USA), MMP9 (1:500, Abcam, Cambridge, UK, able to label both pro-MMP9 and active MMP9), CD31 (1:50, Abclonal, Wuhan, China), and occludin (1:100, Abcam, Cambridge, UK) antibodies. The next day, sections were washed in PBS (3 × 5 min) before being incubated for 2 h in the appropriate secondary antibodies conjugated to Alexa Fluor 488 or 594 (1:1000 Invitrogen, Carlsbad, CA, USA). After another three 5 min washes in PBS, sections were then counterstained with the fluorescent nuclear dye Hoechst 33342 (0.2 mg/100 mL PBS; Sigma, UK) for 5 min to facilitate detection of cell nuclei or in NeuroTrace^®^ 435/455. Slides were coverslipped in ProLong^®^ Gold antifade reagent.

The tyramide signal amplification technique was performed for the detection of AQP4, MMP9 and occludin immunostaining in the cortex. Sections were washed three times (5 min each) in PBS and then incubated with 0.3% hydrogen peroxide for 30 min. After 3 further 5 min washes with PBS, the sections were incubated in rabbit anti-biotin (1:400, Jackson ImmunoResearch, West Grove, PA, USA) for 2 h, washed with PBS, then in avidin-biotin-peroxidase complex (1:250 ‘A’ and 1:250 ‘B’ in PBS, prepared 30 min before use, Vectastain ABC Elite Kit, Vector Laboratories, Burlingame, CA, USA) for 30 min at room temperature. Following 3 × 5 min washes in PBS, sections were incubated with tyramide (1:75, NEN Life Sciences, Boston, MA, USA) for 10 min. After further 3 × 5 min washes in PBS, the sections were incubated with extrAvidin FITC (1:400) for 2 h before the sections were immunostained with another antibody for double-labeling or washed, mounted, and coverslipped.

### 4.7. Histological Analysis

Images around the peri-contusional regions were captured using a fluorescent microscope at ×20 magnification by an experimenter blind to the treatment groups. At least 2 brain sections per animal with 5 regions per section were analyzed. Using ImageJ, the region of interest at 0.25 mm^2^ was selected and the intensity of the staining was obtained as a percentage of the area. To analyze the astrocytic expression of MMP9 and endothelial expression of occludin, a customized macro on ImageJ was used. The glial fibrillary acidic protein (GFAP) immunostaining was initially captured, then the percentage of MMP9 expression within the GFAP immunostaining was calculated. Similarly, the CD31 expression was initially analyzed, then the occludin expression within the CD31 immunostaining was analyzed.

### 4.8. Statistical Analysis

All results were expressed as mean ± standard error of the mean (SEM). The data were tested by parametric analysis of variance (ANOVA). Two-way ANOVA (treatments vs. post-injury time points) was used for behavior test analyses. Correction for multiple comparisons was carried out with Tukey’s post hoc test to determine whether there were differences among groups. The criterion for statistical significance was set at *p* < 0.05 determined by SigmaPlot 12.0.

## Figures and Tables

**Figure 1 ijms-21-06291-f001:**
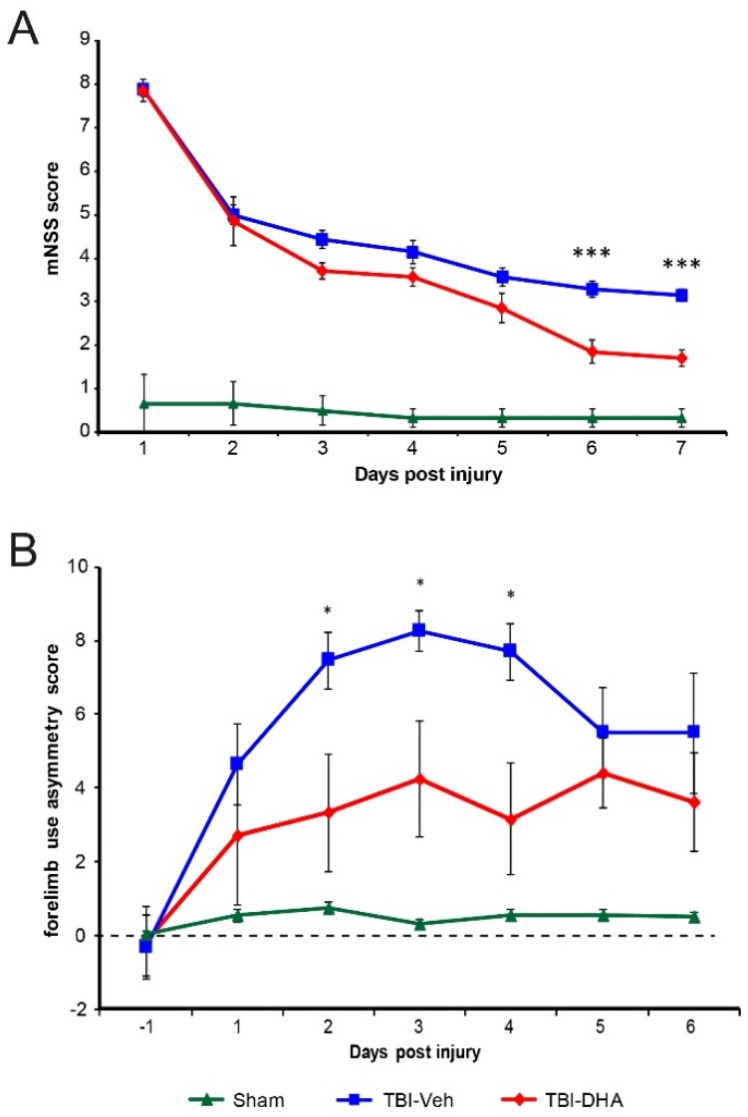
Docosahexaenoic acid (DHA)-treated traumatic brain injury TBI rats exhibited quicker and better recovery than the vehicle-treated TBI rats. (**A**) In the modified Neurological Severity Score (mNSS) test, DHA-treated rats (red diamond, *n* = 8) exhibited improvement in mNSS from day 3 post injury compared to the vehicle-treated TBI rats (blue square, *n* = 8). Sham-operated rats (green triangle, *n* = 7) exhibited limited deficit in the mNSS test throughout the 7 days of testing. (**B**) In the forelimb use asymmetry score test, DHA-treated TBI rats (red diamond, *n* = 8) exhibited improvement in mNSS from day 1 post injury compared to the vehicle-treated TBI rats (blue square, *n* = 8). Sham-operated rats (green triangle, *n* = 7) exhibited a limited difference in forelimb use asymmetry score test throughout the 7 days of testing. The * and *** indicate *p* < 0.05 and *p* < 0.001, respectively.

**Figure 2 ijms-21-06291-f002:**
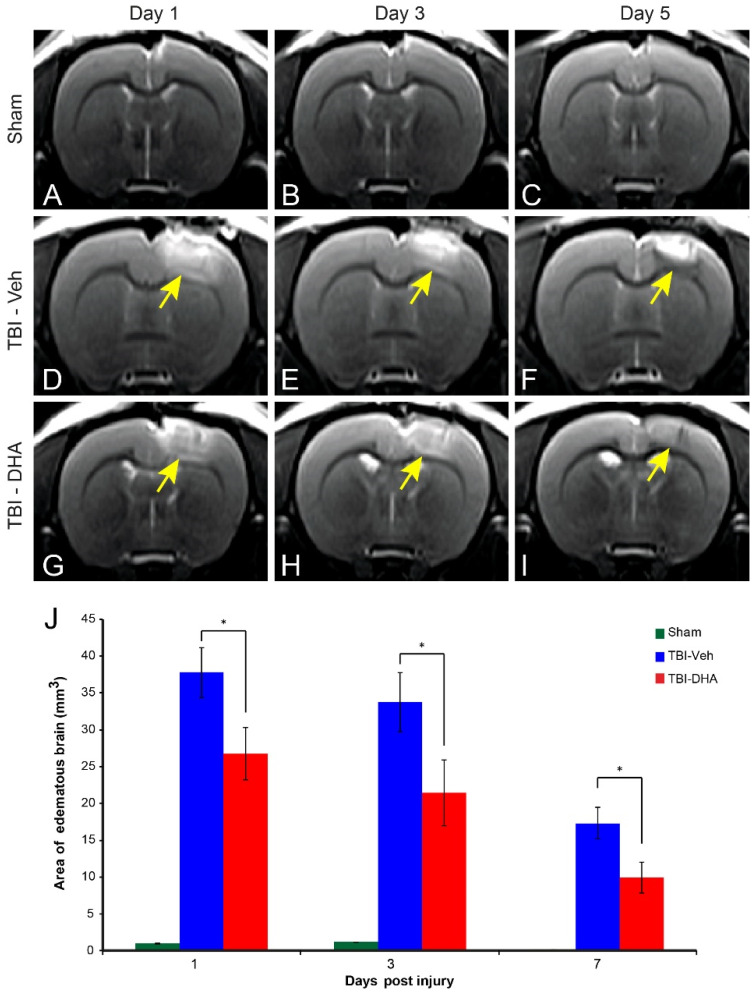
DHA-treated TBI rats exhibited less edema at the injury site than the vehicle-treated TBI rats at various time points. (**A**–**C**) In sham-operated rats, no visible edema was observed at the 3 time points studied. (**D**–**F**) vehicle-treated TBI rats exhibited large edema (yellow arrow) at day 1–7 post injury. (**G**–**I**) DHA-treated TBI rats exhibited some edema (yellow arrow) at day 1 post injury but this reduced with time. (**J**) Analysis of edema at the injury site revealed a significantly reduced edema in DHA-treated rats (red, *n* = 8) compared vehicle-treated TBI rats (blue, *n* = 8) at 1–7 days post injury. Edema in sham-operated rats (green, *n* = 7) was very limited or if any. The * indicates *p* < 0.05.

**Figure 3 ijms-21-06291-f003:**
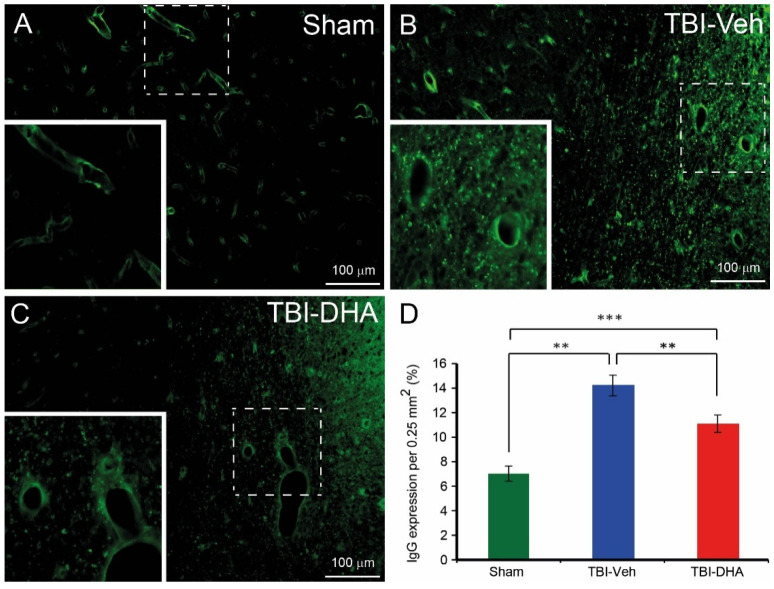
DHA-treated TBI rats exhibited less immunoglobulin G (IgG) extravasation at the peri-injury site than the vehicle-treated TBI rats at 7 days post injury. (**A**) In sham-operated rats, no visible IgG expression was observed outside the blood vessels. (**B**) Vehicle-treated TBI rats exhibited lots of IgG extravasation within the cerebral parenchyma. (**C**) DHA-treated TBI rats exhibited some IgG extravasation within the cerebral parenchyma. (**D**) Analysis of IgG extravasation at the pericontusional site revealed a significantly reduced IgG extravasation in DHA-treated rats (red, *n* = 6) compared vehicle-treated TBI rats (blue, *n* = 6) at 7 days post injury. IgG extravasation in sham-operated rats (green, *n* = 4) was significantly lower than TBI rats. The ** and *** indicate *p* < 0.01 and *p* < 0.001, respectively.

**Figure 4 ijms-21-06291-f004:**
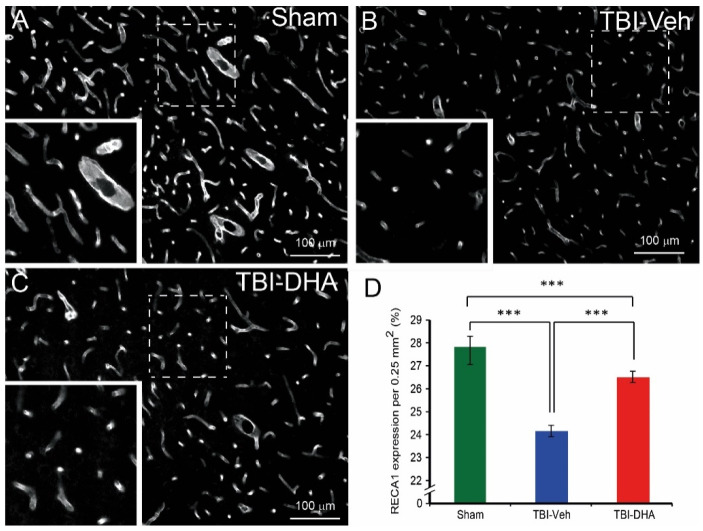
DHA-treated TBI rats exhibited more anti-rat endothelial cell antigen-1 (RECA-1) expression at the peri-injury site than the vehicle-treated TBI rats at 7 days post injury. (**A**) In sham-operated rats, lots of visible RECA-1 expression was observed along the large and small caliber blood vessels. (**B**) Vehicle-treated TBI rats exhibited some RECA-1 expression in small calibre blood vessels. (**C**) DHA-treated TBI rats exhibited some RECA-1 expression in both large and small caliber blood vessels. (**D**) Analysis of RECA-1 expression at the pericontusional site revealed a significantly reduced loss of RECA1 expression in blood vessels in DHA-treated rats (red, *n* = 6) compared vehicle-treated TBI rats (blue, *n* = 6) at 7 days post injury. RECA1 expression in sham-operated rats (green, *n* = 4) was significantly higher than TBI rats. The *** indicates *p* < 0.001.

**Figure 5 ijms-21-06291-f005:**
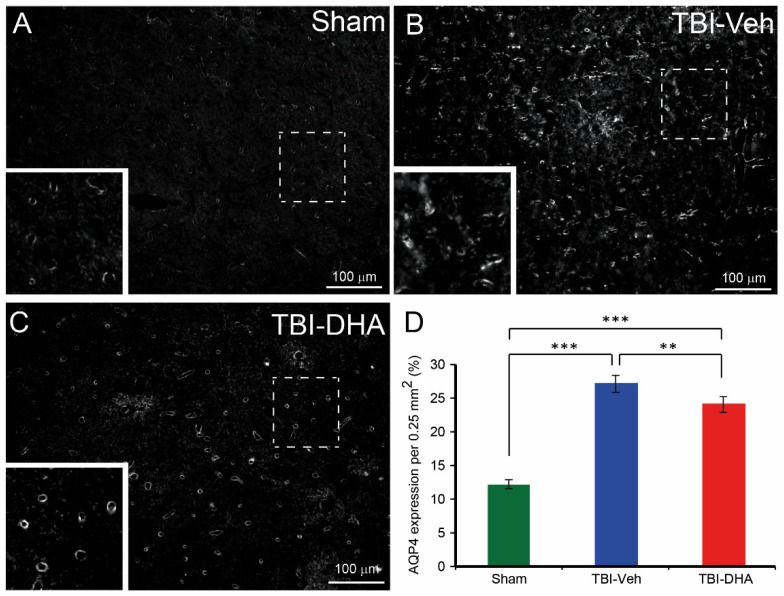
DHA-treated TBI rats exhibited less aquaporin 4 (AQP4) expression at the peri-injury site than the vehicle-treated TBI rats at 7 days post injury. (**A**) In sham-operated rats, some visible AQP4 expression was observed within the blood vessels. (**B**) Vehicle-treated TBI rats exhibited lots of AQP4 expression within and around blood vessels. (**C**) DHA-treated TBI rats exhibited some AQP4 expression within blood vessels. (**D**) Analysis of AQP4 expression at the pericontusional site revealed a significantly reduced loss of AQP4 expression in blood vessels in DHA-treated rats (red, *n* = 6) compared vehicle-treated TBI rats (blue, *n* = 6) at 7 days post injury. AQP4 expression in sham-operated rats (green, *n* = 4) was significantly lower than TBI rats. The ** and *** indicate *p* < 0.01 and *p* < 0.001, respectively.

**Figure 6 ijms-21-06291-f006:**
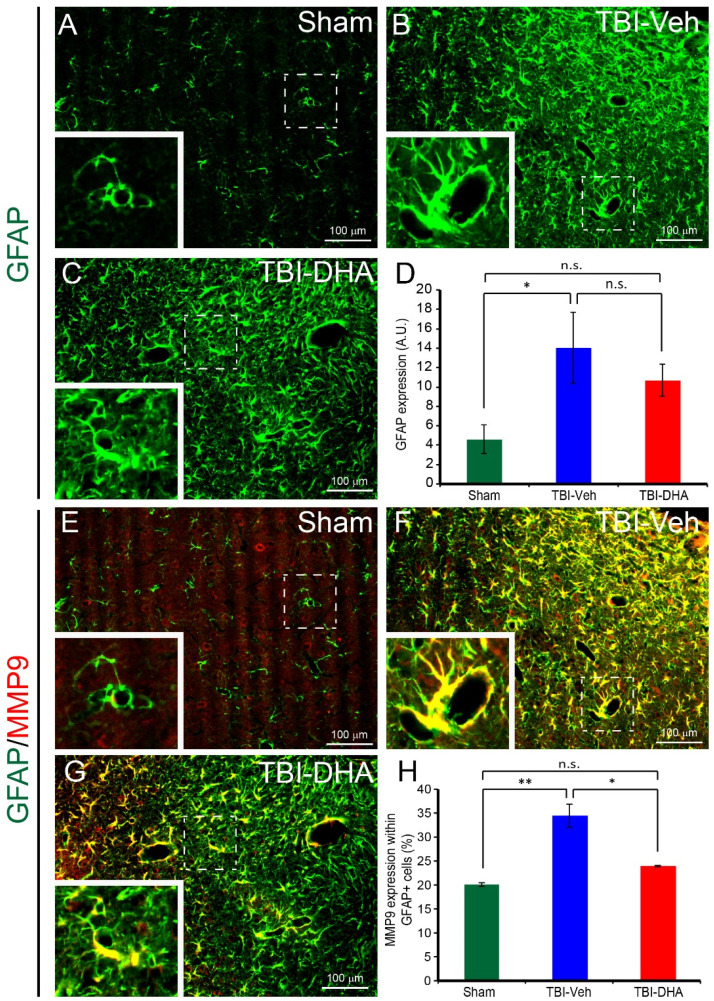
DHA-treated TBI rats exhibited less astrocytic matrix metalloproteinase (MMP9) expression at the peri-injury site than the vehicle-treated TBI rats at 7 days post injury. (**A**,**E**) In sham-operated rats, some visible glial fibrillary acidic protein (GFAP) expression (green) was observed, with many surrounding blood vessels. MMP9 expression (red) was strongly present in neuronal-like cells. (**B**,**F**) Vehicle-treated TBI rats exhibited lots of GFAP expression within the brain parenchyma and around blood vessels. Moreover, MMP9 expression (red) was strongly present in neuronal-like cells and within GFAP immunopositive cells. (**C**,**G**) DHA-treated TBI rats exhibited lots of GFAP expression within the brain parenchyma and around blood vessels. However, MMP9 expression (red) was only strongly present in neuronal-like cells and less within GFAP immunopositive cells. (**D**,**H**) Analysis of GFAP expression or MMP9 within GFAP immunopositive cells at the pericontusional site revealed a significantly reduced loss of astrocytic MMP9 expression in DHA-treated rats (red, *n* = 6) compared vehicle-treated TBI rats (blue, *n* = 6) at 7 days post injury. astrocytic MMP9 expression in sham-operated rats (green, *n* = 4) was significantly lower than in vehicle-treated TBI rats. The * and ** indicate *p* < 0.05 and *p* < 0.01, respectively.

**Figure 7 ijms-21-06291-f007:**
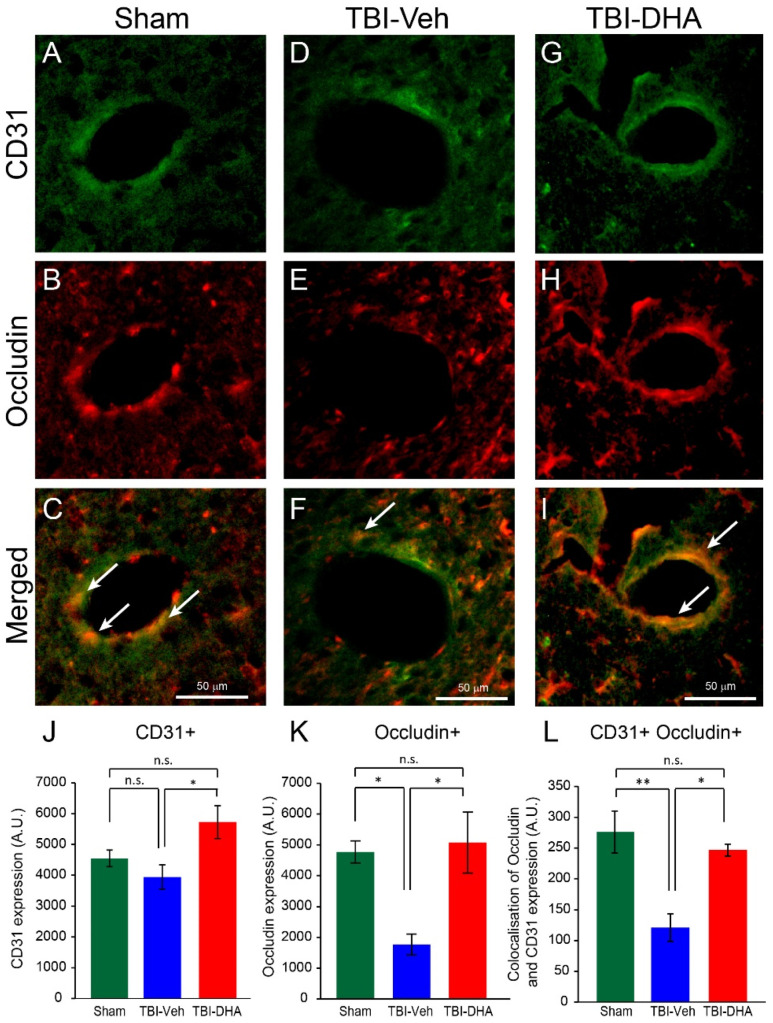
DHA-treated TBI rats exhibited more endothelial occludin expression at the peri-injury site than the vehicle-treated TBI rats at 7 days post injury. (**A**–**C**) In sham-operated rats, CD31 expression (green) was observed strongly with occludin expression (red) in endothelial cells surrounding blood vessels. (**D**–**F**) Vehicle-treated TBI rats exhibited CD31 expression (green) around blood vessels, but the occludin expression (red) was sparsely present in the endothelial cells. (**G**–**I**) DHA-treated TBI rats exhibited CD31 expression (green) around blood vessels and the occludin expression (red) was strongly present in the endothelial cells. (**J**–**L**) Analysis of CD31 expression (**J**) or total occludin expression (**K**) or occludin within CD31 immunopositive endothelial cells (white arrow) (**L**) at the pericontusional site revealed a significantly reduced loss of endothelial occludin expression in DHA-treated rats (red, *n* = 4) compared vehicle-treated TBI rats (blue, *n* = 4) at 7 days post injury. Endothelial occludin expression in sham-operated rats (green, *n* = 4) was significantly higher than vehicle-treated TBI rats. The * and ** indicate *p* < 0.05 and *p* < 0.01, respectively.

**Table 1 ijms-21-06291-t001:** Neurological severity score.

Motor Tests	Points
Raising rat by the tail	3
1 Flexion of forelimb	
1 Flexion of hindlimb	
1 Head moved > 10° to vertical axis within 30 s	
Placing rat on the floor (normal = 0; maximum = 3)	3
0 Normal walk	
1 Inability to walk straight	
2 Circling toward the paretic side	
3 Fall down to the paretic side	
Sensory tests	2
1 Placing balance tests (visual and tactile test)	
2 Proprioceptive test (deep sensation, pushing the paw against the table edge to stimulate limb muscles)	
Beam balance tests (normal = 0; maximum = 6)	6
0 Balances with steady posture	
1 Grasps side of beam	
2 Hugs the beam and one limb falls down from the beam	
3 Hugs the beam and two limbs fall down from the beam, or spins on beam (>60 s)	
4 Attempts to balance on the beam but falls off (>40 s)	
5 Attempts to balance on the beam but falls off (>20 s)	
6 Falls off: No attempt on the balance or hang on to the beam (<20 s)	
Reflexes absent and abnormal movements	4
1 Pinna reflex (head sake when touching the auditory meatus)	
1 Corneal reflex (eye blink when lightly touching the cornea with cotton)	
1 Startle reflex (motor response to a brief noise from snapping a clipboard paper)	
1 Seizure, myoclonus, myodystony	
Maximum points	18

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
