# Peer review of "DHA Attenuates Cerebral Edema Following Traumatic Brain Injury via the Reduction in Blood–Brain Barrier Permeability"

_ijms, 2020, doi:10.3390/ijms21176291_

Round 1
Reviewer 1 Report
All my concerns have been addressed.
Reviewer 2 Report
Authors responded to all concerns
This manuscript is a resubmission of an earlier submission. The following is a list of the peer review reports and author responses from that submission.
Round 1
Reviewer 1 Report
Re: ijms-842243
The manuscript titled “DHA Attenuates Cerebral Edema Following Traumatic Brain Injury via the Reduction in Blood-Brain Barrier Permeability” submitted by Dr. Liu and colleagues evaluates the impact of systemically administered docosahexaenoic acid (DHA) on functional recovery from TBI. The authors of this paper, also showed significant functional recovery of rats from TBI, quantified with the nSS scale and forelimb asymmetry. In the discussion, the researchers postulate the activation of the nuclear factor E2-related factor 2-antioxidant response element (Nrf2-ARE) pathway as a possible mechanism of action, which seems plausible given DHA’s antioxidant properties. In particular, the research begins with the blood brain barrier (BBB), demonstrating that DHA administration reduces BBB leakiness. Proved directly via IgG leakage and edema level, and postulates a mechanism through which this may occur (reduced MMP-9 and AQP4 expression, and a preservation of endothelial RECA-1). Together, this is a well-controlled study, with clear criteria, appropriate tests, clear figures, and a clear conclusion. However, several minor improvements could be made to the discussion:
1.) DHA is found primarily within fatty tissue of the CNS and retina. However, in this paper, the mechanism of action being studied appears to focus exclusively on the endothelial cells of the BBB. Please offer a plausible explanation of how DHA is taken up by endothelial cells.
2.) In the paper, several mechanisms are postulated for the reduction of edema (water entry through AQP-4; reduction in MMP-9 leading to reduced GFAP expression). However, the molecular link between DHA and these findings is not significantly discussed, beyond a cursory mention of TrkB/BDNF. What is the mechanism that the researchers believe that DHA is working through in order to reduce endothelial leakiness?
Reviewer 2 Report
In their manuscript, Liu et al. show a beneficial effect of DHA administration on BBB function after TBI. These are potentially important findings as DHA is safe and can be used clinically in humans. My concerns are as follows:
1) The number of animals (n) analyzed in each experiment should be reported.
2) The reduction seen in RECA + blood vessels may be due to endothelial cell loss, but may also be due simply to a reduced expression in RECA itself. This possibility should be discussed in the manuscript.
3) The AQP4 data is important for reasons mentioned in the discussion. AQP4 however is not a tight junction protein and does not answer the question of whether the tight junctions are altered by TBI or protected by DHA. I recommend staining for a tight junction protein such as ZO-1, occludin, or claudin-5. An alteration in any of these proteins at the tight junction of endothelial cells would support the MRI and IgG extravasation data and make the conclusions more convincing. This is needed to differentiate between changes in paracellular permeability (i.e. changes in tight junction integrity) vs. transcellular permeability (i.e. an increase in transcytosis) at the BBB.
4) It should be stated whether the MMP9 antibody used binds to total MMP9 or activated MMP9.
